# Cardiac Arrest after Small Doses Ropivacaine: Local Anesthetic Systemic Toxicity in the Course of Continuous Femoral Nerve Blockade

**DOI:** 10.3390/ijerph191912223

**Published:** 2022-09-27

**Authors:** Wojciech Gola, Szymon Bialka, Marek Zajac, Hanna Misiolek

**Affiliations:** 1Faculty of Medicine and Health Sciences, Jan Kochanowski University, 25-369 Kielce, Poland; 2Department of Anaesthesiology, Intensive Care and Emergency Medicine, Faculty of Medical Sciences in Zabrze, Medical University of Silesia, 40-055 Katowice, Poland; 3Department of Anesthesiology and Intensive Care, St. Lucas Hospital, 26-200 Konskie, Poland

**Keywords:** local anesthetic systemic toxicity, continuous femoral nerve block, ropivacaine, regional anesthesia, cardiac arrest

## Abstract

Background: The paper presents a case report of an episode of local anesthetic systemic toxicity (LAST) with cardiac arrest after continuous femoral nerve blockade. Case report: A 74-year-old patient burdened with hypertension and osteoarthritis underwent elective total knee replacement surgery. After surgery, a continuous femoral nerve blockade was performed and an infusion of a local anesthetic (LA) was started using an elastomeric pump. Five hours after surgery, the patient had an episode of generalized seizures followed by cardiac arrest. After resuscitation, spontaneous circulation was restored. In the treatment, 20% lipid emulsion was used. On day two of the ICU stay, the patient was fully cardiovascularly and respiratorily stable without neurological deficits and was discharged to the orthopedic department to continue treatment. Conclusion: Systemic toxicity of LA is a serious and potentially fatal complication of the use of LA in clinical practice. It should be noted that in nearly 40% of patients, LAST deviates from the classic and typical course and may have an atypical manifestation, and the first symptoms may appear with a long delay, especially when continuous blockades are used. Therefore, the proper supervision of the patient and the developed procedure in the event of LAST is undoubtedly important here.

## 1. Introduction

Despite the evident and indisputable benefits associated with the use of local anesthetics (LAs), their widespread use in clinical practice and the popularization of regional anesthesiology techniques, one should also remember the potential adverse effects associated with the use of LAs. The toxic effects of LAs relate to both local and systemic toxicity. The systemic toxic effect of LAs may be manifested as neuro- or cardiotoxicity and is the most serious, potentially fatal complication associated with the use of LAs and regional anesthesia [1]. According to the Premier Perspective Database (PPD), the cumulative incidence of LAST is estimated at 0.18%, which is 1.8 LAST cases per 1000 blockades [2]. National survey data from Italy and Finland indicate a significantly lower incidence of LAST at 0.34 and 0.37 cases per 1000 peripheral nerve blocks, respectively [3,4]. However, it must be noted that the actual incidence of LAST is underestimated, mainly due to the lack of reporting of some incidents, inaccuracy of databases and registers or incorrect diagnosis. Although most data indicate a decrease in LAST occurrence in recent years, this problem should not be underestimated. Local anesthetic systemic toxicity is still a complication that occurs even more often than epidural hematoma, or as common as neurological complications of peripheral nerve blockades [1]. In the case of LAST, its classic manifestation is the sequential appearance of symptoms, first from the central nervous system and then from the cardiovascular system. The first phase, in the case of both neuro- and cardiotoxicity, is usually the stimulation phase, followed by depression of the above-mentioned systems [1]. However, it should be remembered that in about 40% of reported LAST cases, the manifestation of clinical symptoms does not follow a classic course with the typical sequential appearance of prodromal symptoms, systemic neurotoxicity and then cardiotoxicity. Almost 50% of the cases of the non-classical LAST occurrence can have a completely atypical course (cardiovascular symptoms without prior manifestation by the central nervous system) or delayed onset of symptoms [1]. In some cases, the first symptoms of LAST may appear 30 or even 60 min after LA injection [5]. This fact is associated with more frequent blockades performed under ultrasound control at present, which reduces the number of unintended direct intravascular LA injections, more frequent use of infiltration techniques and continuous blockades. In addition, even 15% of LAST episodes may be associated with continuous LA infusion (continuous blockades), and the first symptoms are usually observed between days 1 and 4 of the infusion, usually preceded by subtle prodromal symptoms or hemodynamic instability [1]. Therefore, an important element is proper supervision, but also awareness of the patients themselves, which can help in the early identification of the first symptoms of LAST. Finally, early prevention is a key element in reducing the incidence of LAST

Although most of the data show the decreasing frequency of LAST in recent years, one should bear in mind this potentially fatal complication of regional anesthesia. Therefore, the main aim of the study was to update the knowledge about LAST based on the here presented case study.

## 2. Case Presentation

A 74-year-old patient (weight 80 kg, height 162 cm) burdened with hypertension, osteoarthritis and varicose veins in the lower limbs, was scheduled for an elective total right knee replacement surgery (TKR) after a total hip replacement. The patient provided written consent for the publication of this image. The procedure was performed under spinal anesthesia. For neuraxial blockade, a 0.5% solution of isobaric ropivacaine at a dose of 12.5 mg was administered intrathecally. Sedation with a relaxing music (headphones connected to the music player) was used during the procedure. Basic parameters such as non-invasive blood pressure (BP; mmHg), heart rate (HR; bpm), percutaneous saturation (Sp02; %) and body temperature were monitored. During the intraoperative period, no hypotension, episodes of bradycardia or other disturbing symptoms were observed. After the procedure, the patient was transferred to a recovery room where basic hemodynamic parameters (such as BP, HR and Sp02) were monitored. Within the recovery room, a continuous femoral nerve block was performed under ultrasound guidance. After prior aspiration, an induction dose of LA (15 mL 0.375% ropivacaine solution; Molteni, Italy +2.5 µg/mL adrenaline; Polfa Warsaw, Warsaw, Poland) was injected, resulting in the adequate circular, perineural distribution of LA. Then, an elastomeric pump (Easypump II, BBraun, Germany) was connected to the set filled with 270 mL of 0.2% ropivacaine solution with 2.5 µg/mL adrenaline. The rate of infusion was set at 5 mL/h. The patient was discharged to the orthopedic department after a 2 h stay in the recovery room. Within 3 h following the blocking in the orthopedic department, the patient had an episode of generalized seizures. The early warning system team was called. In the ward, the upper respiratory tract was cleared, the oxygen supply through the nasal cannula was turned on and 5 mg of diazepam (Relanium; Polfa Warsaw, Warsaw, Poland) was administered intravenously, resulting in the cessation of the seizures. The infusion of the ropivacaine solution was disconnected and the elastomeric pump was secured for inspection. The subsequent inspection of the pump did not reveal any failure in terms of mechanical damage and the set flow. Due to the suspected neurotoxic effects of LA, the immediate supply of 20% fat emulsion (Intralipid^®^; Fresenius Kabi, Uppsala, Sweden)–100 mL bolus was started. During the assessment of vital signs, bradyarrhythmias and single extrasystoles with broad QRS syndromes were observed, namely Sp02 97%, arterial BP 160/95 mmHg, glucose level-7.9 mmol/L and state of consciousness assessed with GCS-6 points. During the initial assessment of the patient, a renewed episode of generalized seizures occurred followed by sudden cardiac arrest in the ventricular fibrillation mechanism. Resuscitation was undertaken according to ERC (European Resuscitation Council Guidelines 2015) by performing a single defibrillation with a 200 J current and then the return of spontaneous circulation (ROSC) was assessed. During resuscitation, a bolus of 20% fat emulsion was repeated and an Intralipid infusion (200 mL for 20 min) was started. Due to persistent impaired consciousness, the patient required intubation and assisted breathing with an Ambu bag with 100% oxygen substitution. Then, the patient was transferred to the intensive care unit (ICU). As part of the urgent procedure, a computed tomography (CT) of the head was also performed without showing obvious pathologies. In the ICU, the infusion of 20% fat emulsion was continued, and the total dose of Intralipid administered during the stay was 700 mL (10 mL/kg). After the LAST episode, blood samples were not secured for LA concentration measurement due to the lack of such possibility at that time. The patient in the ICU was mechanically ventilated for 120 min. After extubation, no impaired consciousness or seizures were observed. In the first hours of the ICU stay, bradyarrhythmias and atrial fibrillation episodes were observed, and throughout the patient’s stay, the administration of vasoactive and inotropic drugs was not required. Analgesia was carried out using a systemic supply of non-opioid and opioid drug (IV oxycodone continuous infusion, IV paracetamol at 1 g every 6 h, IV metamizole at 1 g). On the second day, the patient, who did not exhibit impaired consciousness and neurological deficits, fully cardiovascularly and respiratorily stable, was transferred to the orthopedics department to continue treatment. During the following days of hospital stay, transthoracic echocardiography (TTE) and Holter ECG was performed. The above tests did not reveal any abnormalities. The neurologic function was intact at the two-week follow-up.

## 3. Discussion

The mechanism of LAST is a complex phenomenon and mainly causes blockades of voltage sensitivity of sodium channels, which leads to conduction inhibition in the central nervous system’s neurons and cardiomyocytes [6]. Other important mechanisms of toxicity are: blockade of voltage sensitivity calcium channels and blockade of sarcoplasmic ryanodine receptors which impairs the release of calcium from intracellular resources [7,8]. Mechanisms of LAST also include: oxidative phosphorylation and Na-K ATPase inhibition which causes impairment of cellular metabolism and ATP production [9,10], blocking the activation of multiple protein kinases (including the serine–threonine kinase Akt, tyrosine kinase Src) [11], inhibition of adenylate cyclase as well as adrenoreceptors [6]. The first experimental study in a rat model that suggested efficiency of lipid emulsion infusion in LAST was reported by Weinberg in 1998 [12]. In 2006, Rosenblatt et al. reported the successful use of a 20% lipid emulsion in the resuscitation of a patient after a bupivacaine-related cardiac arrest [13]. The most frequent postulated mechanism of action of lipid emulsion was “lipid sink/sponge” which puts forward the idea that lipid emulsion, similar to a sponge, adsorbs highly lipophilic substances (LAs) and reduces the serum concentration [14]. As a result, the distribution of LA is limited while redistribution increases from tissue to plasma. Current studies and reports support a slightly broader concept of the mechanism of action of the lipid emulsion, which is explained as a dynamic shuttling effect for lipophilic substances. Lipid emulsion could be an active transporter of the LA from the key organs with high blood flow (brain, heart) to those less sensitive to the action of xenobiotics (muscles, liver), where some of them are also metabolized [15,16,17]. Other mechanisms of action are also postulated. Lipid emulsion infusion may increase cardiac contractility as well as cardiac output and flow thought the CNS. The proposed mechanism of the above-described is due to simple positive inotropic effect and increase in preload related to volumetric resuscitation with lipid emulsion. Additionally, the infusion of the lipid emulsion also increases systemic pressure by a direct effect on the peripheral vessels. The above mechanism is not yet fully understood [18]. The mechanisms of action of lipid emulsion also include the activation of cardioprotective mechanisms [4].

In the analyzed case, the blockade was performed with ultrasound, which, according to current data, contributes to a nearly 65% reduction in LAST risk [19]. Nevertheless, it should be remembered that despite a significant risk reduction, this potentially fatal complication, as the analyzed case shows, is still a clinically significant problem. As a result, each Department of Anesthesiology should develop a protocol in the event of LAST occurrence and a 20% fat emulsion (LAST Rescue Kit) should be available. According to current guidelines, in the case of the first serious symptoms of LA systemic toxicity, early implementation of therapy with a 20% fat emulsion should be considered, with the early administration of Intralipid being more important than its precise dosage [1]. In the event of sudden cardiac arrest as an effect of LA cardiotoxicity, it is also recommended to promptly notify the nearest cardiopulmonary bypass team due to the possibility of prolonged resuscitation. After the LAST episode, proper supervision and monitoring of the patient is also a key element. According to the American Society of Regional Anesthesia guidelines, the patient should be monitored for a minimum of 2 h for an isolated neurological incident, and in the event of a cardiovascular incident as a manifestation of LAST, the monitoring of the patient should be extended to a minimum of 4–6 h [1].

The following situations should be considered in the differential diagnosis of LAST: cardiogenic cause of cardiac arrest, neurological event followed by cardiac arrest, anaphylaxis and anaphylactic shock. Both cardiogenic and neurological were ruled out as causes of cardiac arrest (CT, TEE, Holter ECG revealed no abnormalities). Anaphylaxis as a potential cause of cardiac arrest should also be considered, and additionally the level of serum tryptase should be measured [20]. In the presented case, such diagnosis was not performed due to the typical clinical picture of LAST and an adequate response to the applied treatment. Plasma LA concentrations may also bring valuable information in the diagnosis of LAST but requires rigorous sample protocols and its analysis which may not be obtained in all cases [21]. In the presented case, such a measurement could not be performed then, and the department did not conduct appropriate procedures.

When an elastomeric pump is used, its dysfunction should also be considered as a possible cause of the adverse event [22]. In the presented case report, the elastomeric pump was inspected for failures, but the inspection of the pump did not reveal any failure in terms of mechanical damage and the set flow. The set flow in the elastomeric pump may also depend on various external factors (temperature, pressure), which may ultimately affect the dose of the administered drug [23]. In most cases, it does not affect peripheral blockades but may be crucial, e.g., in epidural anesthesia.

## 4. Conclusions

We present the case of a patient who was successfully treated after a cardiac arrest secondary to LAST from ropivacaine in a small dose (cumulative dose of 106.25 mg/1.328 mg/kg body weight within 5 h) used as analgesia after total knee replacement surgery. While reports of LAST are rare, cardiac arrest associated with toxicity may happen after small doses of LA. Therefore, the constant monitoring of the patients and a prepared strategy of management in case of LA toxicity symptoms are necessary.

## Data Availability

The datasets used and/or analyzed during the current study are available from the corresponding author upon reasonable request. These data are completely de-identified. No individual personal data, in any format, is reported.

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
