# Peer review of "Cardiac Arrest after Small Doses Ropivacaine: Local Anesthetic Systemic Toxicity in the Course of Continuous Femoral Nerve Blockade"

_ijerph, 2022, doi:10.3390/ijerph191912223_

Round 1
Reviewer 1 Report
Local anesthetic systemic toxicity is an important topic. The authors presented the case in an interesting way. However, some points need to be fixed to fit for publication as follows:
1. Author names “Wojciech Gola1 , *, Szymon Bialka 2 , Marek Zajac3 , Hanna Misiolek2 and” is there is missed author names?. Please, revise.
2. There is a problem in using abbreviations throughout the manuscript. The full term should be mentioned first with the abbreviation between paresis then the abbreviations should be exclusively used throughout the manuscript. E.g., in line 31 LA should be written as a local anesthetic (LA) then the abbreviation should be exclusively used further. Such errors have been repeated for many abbreviations throughout the manuscript.
3. The introduction is very concise and the authors used one reference at the end of the two paragraphs (Neal et al 2018). Most of the information present in the discussion (lines 108-133) should be transferred to the introduction.
4. The authors need to update their references with studies in 2021 and 2022.
5. It is not preferable to begin sentences with abbreviations like that in line 41 “LAST is still a complication…..”.and line 47 “TKR was performed …..”
6. The discussion needs to deepen on the mechanism of toxicity, generalized seizures pathogenesis, mechanism of action of therapies used, …etc”. Transfer the general information to the introduction.
Author Response
Dear Reviewer,
First of all, thank you very much for the review of the manuscript and the attached comments.
Please find our replay below:
- Author names “Wojciech Gola1 , *, Szymon Bialka 2 , Marek Zajac3 , Hanna Misiolek2 and” is there is missed author names?. Please, revise.
Author's reply: Author names are correct, names are also entered.
- There is a problem in using abbreviations throughout the manuscript. The full term should be mentioned first with the abbreviation between paresis then the abbreviations should be exclusively used throughout the manuscript. E.g., in line 31 LA should be written as a local anesthetic (LA) then the abbreviation should be exclusively used further. Such errors have been repeated for many abbreviations throughout the manuscript.
Author's reply: Changes have been made according to your comments.
- The introduction is very concise and the authors used one reference at the end of the two paragraphs (Neal et al 2018). Most of the information present in the discussion (lines 108-133) should be transferred to the introduction.
Author's reply: Changes have been made according to your comments.
- The authors need to update their references with studies in 2021 and 2022.
Author's reply: References have been updated according to your suggestions.
- It is not preferable to begin sentences with abbreviations like that in line 41 “LAST is still a complication…..”.and line 47 “TKR was performed …..”
Author's reply: Changes have been made according to your comments.
- The discussion needs to deepen on the mechanism of toxicity, generalized seizures pathogenesis, mechanism of action of therapies used, …etc”. Transfer the general information to the introduction.
Author's reply: Changes have been made according to your comments.
Reviewer 2 Report
Cardiac arrest from local anesthetics is a very interesting topic. There are not many articles about it, interesting the manuscript of Braley et al. which should therefore be cited (Bradley Stauber, B. S., Lin Ma, M. D., & Reza Nazari, M. D. (2012). Cardiopulmonary arrest following cervical epidural injection. Pain Physician, 15, 147-152). However, despite the peculiarity of the manuscript, there are numerous elements to be clarified:
1) How do the authors exclude even a late anaphylactic shock? Authors should respond by citing this article as well (Esposito, M., Montana, A., Liberto, A., Filetti, V., Nunno, N.D., Amico, F., Salerno, M., Loreto, C. and Sessa, F., 2021, January. Anaphylactic Death: A New Forensic Workflow for Diagnosis. In Healthcare (Vol. 9, No. 2, p. 117). MDPI.)
2) The case is well presented however in the introduction the authors should better clarify the purpose of the study.
3) The discussion and conclusions are very meager, they should be expanded and the authors should try to hypothesize a pathophysiological mechanism underlying cardiac arrest. Thus it would appear a diagnosis of exclusion in the absence of evidence.
4) The references within the text do not follow the paper template.
Author Response
Dear Reviewer,
First of all, thank you very much for the review of the manuscript and the attached comments.
Please find our replay below:
1) How do the authors exclude even a late anaphylactic shock? Authors should respond by citing this article as well (Esposito, M., Montana, A., Liberto, A., Filetti, V., Nunno, N.D., Amico, F., Salerno, M., Loreto, C. and Sessa, F., 2021, January. Anaphylactic Death: A New Forensic Workflow for Diagnosis. In Healthcare (Vol. 9, No. 2, p. 117). MDPI.)
Author's reply: Due to the typical clinical picture strongly suggesting LAST and the adequate response to the treatment applied, the diagnosis of anaphylactic shock was not undertaken in presented case report - information supplemented in the manuscript. Information on anaphylaxis was added to the manuscript as an element for differential diagnosis. The citation was also included in the references.
2) The case is well presented however in the introduction the authors should better clarify the purpose of the study
Author's reply: The introduction has been changed to better explain the aims of the study
3) The discussion and conclusions are very meager, they should be expanded and the authors should try to hypothesize a pathophysiological mechanism underlying cardiac arrest. Thus it would appear a diagnosis of exclusion in the absence of evidence.
Author's reply: Both parts of the manuscript have been supplemented according to your suggestions
4) The references within the text do not follow the paper template
Author's reply: The references have been changed.
Reviewer 3 Report
Review of a case report on “Cardiac arrest after small doses ropivacaine: LAST in the course of continuous femoral nerve blockade”.
Although not unknown, this case report shows a possible draw-back of local anesthesia techniques in current clinical practice.
A few points raise reading this manuscript:
1) Lacking is the permission of the patient for publishing this report.
2 2) The case presentation can and should be condensed
a. Line 49-50 English style can be improved
b. Placement of the femoral catheter with all ultrasonographic info is provided, but most is not very relevant for this case report; condense / delete lines 57-71.
3 3) Add information on how analgesia was handled in this case after the resuscitation. Lines 102-103 are not specific enough on this point, at present. This item can also be further discussed in the discussion section.
4 4) Were blood-samples drawn to analyze concentration of local anesthetics? If yes, provide data, if no give reasoning why not done. When LA-concentration can be added, the case and claim “after small doses” can be supported.
5 5) Was the functioning of the elastomeric pump checked after this incident? Please provide details and discuss this point in the discussion section also. At present the rate of infusion was provided to be set and given at 5 ml/h (line 76), but in case of failure of the device it might have been much more, giving a sudden toxic reaction, as the pump was filled with 270 ml of LA. Add pro’s and con’s of using such a device; this might give extra, for the reader interesting, information in this case. Furthermore, in case the device indeed has failed, the claim on “small doses ropivacaine” being responsible for this accident must be reviewed.
6 6) Was echocardiography performed in this patient? It is a 74-yrs-old person with hypertension, and may be unknown for cardiovascular disease, having a period of resuscitation after an operation. Although the resuscitation-period started with seizure, which makes an effect because of LAST most likely, other etiologies deserve to be addressed as well, amongst being cardiovascular disease. Please add some remarks to the discussion section.
7 7) The English style used can be improved, by e.g. a native English speaker, which will improve readability of this case report a little bit.
Author Response
Dear Reviewer,
First of all, thank you very much for the review of the manuscript and the attached comments.
Please find our response below:
- Lacking is the permission of the patient for publishing this report:
Author's reply: Patient provided written consent for publication of this image, this has been also completed in the manuscript
2 2) The case presentation can and should be condensed
- Line 49-50 English style can be improved:
Author's reply: English style has been improved
- Placement of the femoral catheter with all ultrasonographic info is provided, but most is not very relevant for this case report; condense / delete lines 57-71.
Author's reply: This section has been condensed
3 Add information on how analgesia was handled in this case after the resuscitation. Lines 102-103 are not specific enough on this point, at present. This item can also be further discussed in the discussion section.
Author's reply: This section has been supplemented with the suggested information.
4 Were blood-samples drawn to analyze concentration of local anesthetics? If yes, provide data, if no give reasoning why not done. When LA-concentration can be added, the case and claim “after small doses” can be supported.
Author's reply: The LA serum concentration level would be a valuable supplement to the case report. Unfortunately was not measured due to the lack of this possibility at that time. Claim “after small doses” refers in this case to the actual dose of the drug administered peripherally. A comment on this was added in the manuscript.
5 Was the functioning of the elastomeric pump checked after this incident? Please provide details and discuss this point in the discussion section also. At present the rate of infusion was provided to be set and given at 5 ml/h (line 76), but in case of failure of the device it might have been much more, giving a sudden toxic reaction, as the pump was filled with 270 ml of LA. Add pro’s and con’s of using such a device; this might give extra, for the reader interesting, information in this case. Furthermore, in case the device indeed has failed, the claim on “small doses ropivacaine” being responsible for this accident must be reviewed.
Author's reply: After cardiac arrest the elastomeric pomp was secured for inspection. Subsequent inspection of the pump did not reveal any dysfunctions in terms of mechanical damage or flow settings. Relevant comments on this subject have been also added in the text.
6 Was echocardiography performed in this patient? It is a 74-yrs-old person with hypertension, and may be unknown for cardiovascular disease, having a period of resuscitation after an operation. Although the resuscitation-period started with seizure, which makes an effect because of LAST most likely, other etiologies deserve to be addressed as well, amongst being cardiovascular disease. Please add some remarks to the discussion section.
Author's reply: During hospitalization transthoracic echocardiography and Holter EKG were performed. The above diagnostic tests did not reveal any abnormalities. Information about that has been added in the manuscript
7 The English style used can be improved, by e.g. a native English speaker, which will improve readability of this case report a little bit.
Author's reply: The manuscript was also revised by a native English speaker.
Round 2
Reviewer 1 Report
No further comments to be addressed
Author Response
Dear reviewer Thank you very much for the effort you put into the review of the manuscript, all comments and favorable opinion
Reviewer 2 Report
The manuscript has been better structured and is ready for publication.
Author Response
Dear Reviewer Thank you very much for the effort you put into the review of the manuscript, all comments and favorable opinionReviewer 3 Report
Review on manuscript: cardiac arrest after small doses ropivacaine: LAST in the course of continuous femoral nerve blockade.
With interest I have read the revised manuscript, which has now clearly improved.
Only one minor point and some textual items rest.
Minor point: line 215: Do the authors indeed mean that patients should be transferred to a heart center for cardiopulmonary bypass support, or is the intended mean to warn the nearest resuscitation team?
Textual:
Line 58: “It is also worth noting that”, can be deleted
Line 60: “at the moment”, should be “at present”
Line 69: frequency of the last: please delete “the”
Line 72: “based on presented case study”: please change into “based on the here presented case study”
Line 167: “et all”, should be “et al”
Line170: “which can be explain that lipid emulsion”, change into “which explains that lipid emulsion”
Line 171: change “reducing” into “reduces” and change “them” into “the”
Line 174: change “which can be explain as” into “explained as”
Line 178: “infusion can increase” into “infusion may increase”
Line 182: “emulsion can also increase” into “emulsion will also increase” and “pressure by direct effect” into “pressure by a direct effect
Author Response
Dear Reviewer,
First of all, thank you for the re-review of the manuscript, and all your comments.
Please find our replay below:
Minor point: line 215: Do the authors indeed mean that patients should be transferred to a heart center for cardiopulmonary bypass support, or is the intended mean to warn the nearest resuscitation team?
Author's reply: According to ASRA Guidelines, it is recommended to inform (warn) the nearest ECMO Center (or the ECMO Team in a given hospital) in advance due to possibility of prolonged resuscitation and the high probability of implementation VA ECMO therapy . VA ECMO therapy gives us time to dissociate of LA from sodium channels and drug metabolism (which may take several hours in some cases) and finally increases the chance of ROSC.
Textual:
Line 58: “It is also worth noting that”, can be deleted
Author's reply: Line: It is also worth noting that”, has been deleted
Line 60: “at the moment”, should be “at present”
Author's reply: “at the moment” has been changed to “at present”
Line 69: frequency of the last: please delete “the”
Author's reply: It has been delated
Line 72: “based on presented case study”: please change into “based on the here presented case study”
Author's reply: "here" has been added
Line 167: “et all”, should be “et al”
Author's reply: it has been changed
Line170: “which can be explain that lipid emulsion”, change into “which explains that lipid emulsion”
Author's reply: it has been changed
Line 171: change “reducing” into “reduces” and change “them” into “the”
Author's reply: “reducing” has been changed into “reduces” and “them” into “the”
Line 174: change “which can be explain as” into “explained as”
Author's reply: “which can be explain as” has been changed into “explained as”
Line 178: “infusion can increase” into “infusion may increase”
Author's reply: “infusion can increase” has been changed into “infusion may increase”
Line 182: “emulsion can also increase” into “emulsion will also increase” and “pressure by direct effect” into “pressure by a direct effect
Author's reply: it has been changed